# BTReport: A Framework for Brain Tumor Radiology Report Generation with Clinically Relevant Features

Juampablo E. Heras Rivera[*][1]  [iD]       JEHR@UW.EDU

Dickson T. Chen[*][1]  [iD]       DTCHEN19@UW.EDU

Tianyi Ren[1]  [iD]       TR1@UW.EDU

Daniel K. Low[2]  [iD]       DALOW@UW.EDU

Jacob Ruzevick[2]       RUZEVICK@UW.EDU

Asma Ben Abacha[3]  [iD]       ABENABACHA@MICROSOFT.COM

Alberto Santamaria-Pang[3],[4]  [iD]       ALBERTO.SANTAMARIAPANG@MICROSOFT.COM

Mehmet Kurt[1]  [iD]       MKURT@UW.EDU

[1] *University of Washington*

[2] *University of Washington School of Medicine*

[3] *Microsoft Health AI*

[4] *The Johns Hopkins University School of Medicine*

**Editors:** Accepted for publication at MIDL 2026

## Abstract

Recent advances in radiology report generation (RRG) have been driven by large paired image-text datasets; however, progress in neuro-oncology RRG has been limited due to a scarcity in open paired image-report datasets. Here, we introduce BTReport, an open-source framework for brain tumor RRG that constructs natural language radiology reports using reliably extracted quantitative imaging features. Unlike existing approaches that rely on general-purpose or fine-tuned vision-language models for both image interpretation and report composition, BTReport performs deterministic feature extraction of clinically-relevant features, then uses large language models only for syntactic structuring and narrative synthesis. By separating RRG into deterministic feature extraction and report generation stages, synthetically generated reports are completely interpretable and contain reliable numerical measurements, a key component lacking in existing RRG frameworks. We validate the clinical relevance of BTReport-derived features, and demonstrate that BTReport-generated reports more closely resemble reference clinical reports when compared to existing baseline RRG methods. To further research in neuro-oncology RRG, we introduce BTReport-BraTS, a companion dataset that augments BraTS imaging with synthetic radiology reports generated with BTReport, and BTReview, a web-based platform for validating the clinical quality of synthetically generated radiology reports. Code for this project can be found at: https://github.com/KurtLabUW/BTReport.

**Keywords:** Brain MRI, Radiology report generation, VASARI, Midline shift, Open dataset, Multimodal learning, Neuro-oncology

## 1. Introduction

Radiology is a medical specialty that employs a variety of imaging modalities (e.g., X-ray, computed tomography (CT), multi-parametric magnetic resonance imaging (mpMRI) for

---

[*] Contributed equally

the detection and monitoring of human disease. The radiology report contains a detailed summary of imaging findings, providing insights into a patient's condition crucial for diagnosis and clinical decision making. With a growing aging population, the demand for radiology services is expected to increase between 16.9% to 26.9% by 2055, while attrition in radiology also continues to increase (Christensen et al., 2025). As the gap between physician workload and available workforce widens, assisted radiology report generation (RRG) is positioned to help address this unmet clinical need.

RRG leverages advances in artificial intelligence (AI) to extract quantitative imaging markers from raw unstructured data in an automated manner. In clinical workflows, RRG promises to improve data quality, repeatability, factual completeness, and timeliness of radiology reporting. The adoption of vision-language models (VLMs) in RRG has led to substantial advances, allowing models to jointly reason over medical images and textual findings (Liu et al., 2019; Chen et al., 2020; Sellergren et al., 2025). These developments have been primarily driven by large-scale chest X-ray datasets such as MIMIC-CXR (Johnson et al., 2019) and IU X-ray (Demner-Fushman et al., 2016), which provide over 300,000 chest X-ray images paired with clinical reports. However, accessible image-report datasets across other radiology specialties are limited. For instance, in neuro-oncology, large neuroimaging datasets are made openly available through efforts such as BraTS (de Verdier et al., 2024), but paired text reports for VLM training remain lacking.

Advances in computer vision have automated glioblastoma (GBM) segmentation from mpMRI, enabling reliable quantification of the contrast-enhancing lesion, necrotic core, and peritumoral edema (Menze et al., 2014). However, segmentations alone lack the clinical context needed to evaluate the impact of tumor on the surrounding brain environment. Neuro-radiologists supplement their interpretation with additional imaging-derived evidence, such as subregion involvement and midline shift (MLS) measurements, among others. Together, these descriptors provide comprehensive brain tumor characterization, and improve risk stratification, treatment planning, and outcome prediction. Neuro-oncology RRG models should reliably include these features to more closely align with radiologist expectations.

Here we introduce BTReport, a two-stage framework for neuro-oncology RRG grounded in clinically relevant quantitative features. BTReport first deterministically extracts imaging markers from mpMRI including patient metadata, VASARI (Visually AcceSAble Rembrandt Images) features, and automated 3D midline shift measurement. These markers are then provided as structured inputs to large language models (LLMs) for clinical reasoning and synthetic report generation. Because quantification is performed algorithmically upstream, BTReport enables measurement-grounded reporting without requiring task-specific fine-tuning of vision encoders or VLMs. This design addresses a key challenge in neuro-oncology RRG: clinical reports frequently include quantitative measurements such as lesion size, volume, and midline shift, which may not be reliably extracted by generic vision encoders without explicit training for measurement and precise spatial reasoning (Chen et al., 2024). By grounding report generation in deterministically extracted imaging features, our framework is directly interpretable and reduces the likelihood of critical detail omission (Wu et al., 2025).

Our contributions are as follows: **a)** a scalable brain tumor mpMRI radiology report generation framework driven by deterministically extracted neuroimaging features (Section 4.1), **b)** a robust, interpretable 3D midline shift (MLS) estimation algorithm (Section 4.3),

**c)** clinical validation of BTReport-derived features demonstrated with semantic clustering of radiology concepts from reference reports, and retrospective modeling of overall survival (Section 4.5.2), and **d)** release of BTReport-BraTS, an open-source image-report companion dataset augmenting BraTS cases with clinically grounded anatomical and pathological descriptors (Appendix G).

## 2. Related Work

A variety of approaches have been proposed for the task of image-paired RRG and generally fall into one- and two-stage frameworks. The leading paradigm for RRG involves training monolithic VLM foundation models to extract image features and generate reports in one step, such as in MedGemma (Sellergren et al., 2025) and MedPaLM-2 (Singhal et al., 2023). Approaches in neuro-oncology following this paradigm include TextBraTS (Shi et al., 2025), which directly prompts GPT-4 models (Achiam et al., 2023) with videos of 2D mpMRI axial slices alongside tumor segmentation masks derived from FLAIR imaging. Radiologists refine these annotations into structured textual labels, which are then used to guide tumor segmentation models toward clinically relevant regions, improving segmentation performance. Although these annotations improved segmentation performance, they do not contain quantitative measurements typically reported by radiologists.

Other image-paired RRG frameworks adopt a two-stage design. For example, *From Segmentation to Explanation* (*Seg-to-Exp*) (Valerio et al., 2025) first establishes descriptive tumor-ROI relationships based on the co-registration of tumor segmentation masks to an anatomical atlas. These relationships are extracted as structured features that describe the percent overlap of the brain tumor with ROIs it occupies, and then provided to an LLM for report generation. The generated reports provide details about regional tumor impact and offer insightful clinical implications based on the functional role associated with identified tumor-ROI relationships. However, generated reports do not resemble the standard narrative structuring of radiology reports authored by fellowship trained neuro-radiologists. Additionally, *Seg-to-Exp* reports do not describe additional clinically-relevant imaging observations beyond anatomical tumor location.

*AutoRG-Brain* (Lei et al., 2024) also adopts a two-stage framework. In the first stage, anomaly ROI masks are generated and mapped to specific ROIs based on anatomical segmentations. In the second stage, a fine-tuned VLM takes embedded anomaly ROI masks as visual prompts for RRG. This approach uses deterministic features to focus the VLM, generating region-specific findings grounded in areas of the image where anomalies were detected. Based on this design, generated reports are largely descriptive, with findings expressed through signed magnitude operators (e.g., "enlarged", "high signal") that encode comparative attribute relationships between detected anomalies and their anatomical location. Importantly, describing an imaging abnormality in relation to anatomical context does not, by itself, establish the clinical context required for higher-order clinical interpretation. As a result, observations from generated reports are not framed in a clinically meaningful way, limiting their utility in informing downstream clinical decision making. Furthermore, reference reports used for fine-tuning contain a limited set of clinically-relevant features.

The framework proposed in *RadGPT* (Bassi et al., 2025) first deterministically extracts clinically-relevant features from CT scans of abdominal tumors,then uses LLMs for syntactic

structuring. This approach produces radiology reports that more closely resemble reference reports when compared to end-to-end report generation models. However, it remains unclear whether a framework similar to *RadGPT* can be applied for neuro-oncology RRG based on deterministically extracted features from mpMRI.

## 3. Data

This study uses two complementary datasets to support radiology report generation and validation of BTReport-derived features. First, the HuskyBrain dataset is a retrospective cohort of fully de-identified pre-operative mpMRI studies paired with radiologist-authored reports, serving as the primary resource for training and evaluation of neuro-oncology RRG frameworks. The second dataset is a subset of BraTS'23 cases, which enable survival analyses with imaging and clinical metadata. Together, these datasets support the evaluation of synthetic report clinical quality and the use of deterministic neuroimaging features for survival outcome prediction.

### 3.1. HuskyBrain Dataset

We collected pre-operative mpMRI scans and radiology reports from a retrospective cohort of GBM patients (n=184) treated at the University of Washington Medical Center (UWMC), an academic medical institution serving patients across the WWAMI region (Washington, Wyoming, Alaska, Montana, and Idaho). Inclusion criteria were: 1) confirmed histopathologic diagnosis of GBM; 2) availability of pre-operative mpMRI sequences including T1, T2, T1c, and T2-FLAIR; and 3) a corresponding pre-operative diagnostic radiology report. Images were pre-processed using CaPTk (Davatzikos et al., 2018; Pati et al., 2020), following BraTS 2017-2023 procedures: DICOM to NIfTI conversion, SRI24 co-registration, 1 mm isotropic resampling, and skull-stripping. Reference reports were authored by fellowship-trained radiologists and selected as the clinical reference standard. For each case, the HuskyBrain dataset contains de-identified MRI sequences, radiologist-authored reports stripped of protected health information, and tumor masks.

### 3.2. Survival Analysis Dataset

From the BraTS'23 dataset (Adewole et al., 2023), a smaller cohort of mpMRI cases (n=461) was used for survival analyses. Cases were selected based on the availability of five minimum metadata entries: (1) age at initial diagnosis, (2) biological sex, (3) confirmed methylation status of O6-methylguanine-DNA methyltransferase (*MGMT*), (4) mutation status of isocitrate dehydrogenase 1 (*IDH1*), and (5) overall survival or equivalent survival quantification representing the number of days between radiological diagnosis and reported days to known death. We collected these demographic and genomic features from multiple publicly available collections of GBM cases, including those from the University of California San Francisco (UCSF-PDGM) (Calabrese et al., 2022), University of Pennsylvania (UPenn-GBM) (Bakas et al., 2021), Clinical Proteomic Tumor Analysis Consortium (CPTAC-GBM) (National Cancer Institute Clinical Proteomic Tumor Analysis Consortium (CPTAC), 2018), and The Cancer Genome Atlas from the Cancer Imaging Archive (TCGA-LGG & TGCA-GBM) (Pedano et al., 2016) (Scarpace et al., 2016).

## 4. Methods

### 4.1. BTReport Framework

BTReport is a two-stage approach for neuro-oncology RRG that first deterministically extracts interpretable and clinically relevant neuroimaging features from mpMRI, then uses an LLM for syntactically structured narrative construction and report generation. Since quantitative measurements are central to radiological evaluation, BTReport relies on validated open-sourced algorithms to derive descriptors of anatomy and tumor pathology, rather than performing end-to-end medical inference with a VLM.

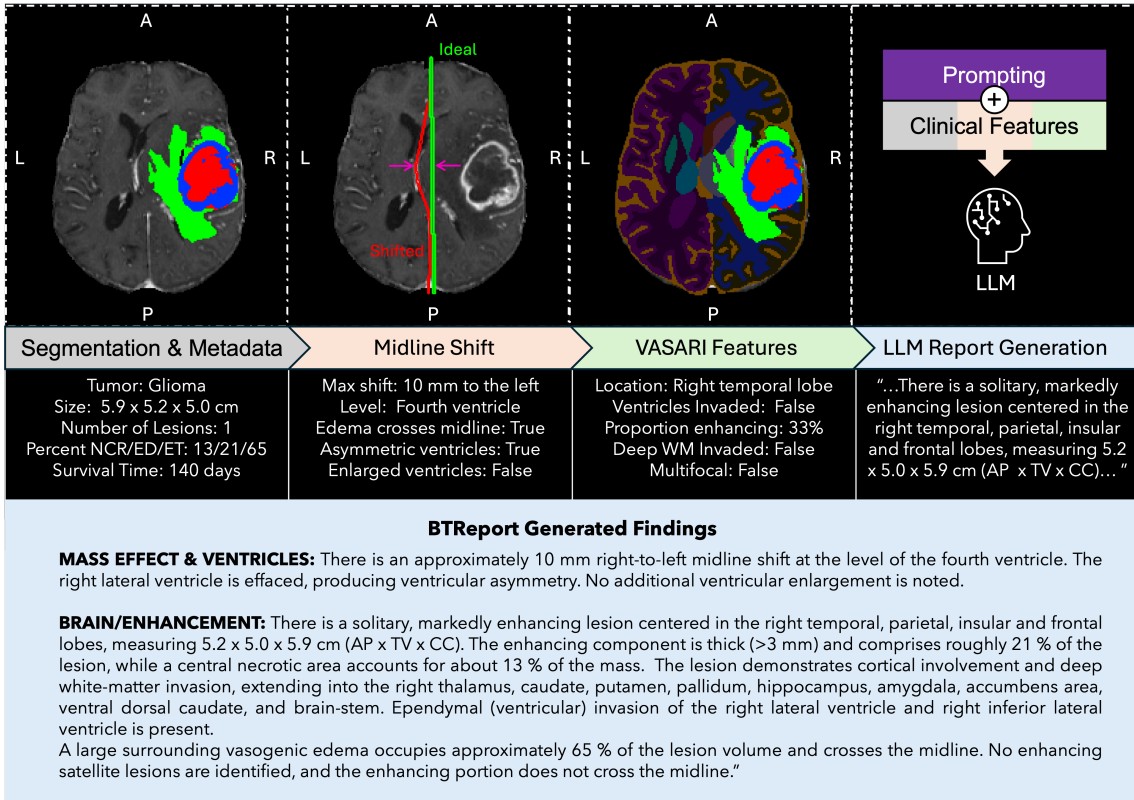

Figure 1: BTReport Overview: Interpretable, clinically meaningful variables are deterministically extracted for each case, including demographics, VASARI features, and 3D midline shift measurements. These features are utilized by context-guided LLMs for clinically grounded radiology report generation.

For a given case, the BTReport framework (Figure 1) consists of four steps:

1. T1-weighted scans and tumor masks are used to generate tumor-robust anatomical segmentations, from which regional volumetric statistics are computed (Section 4.2).
2. Midline shift is quantified by propagating a hand-annotated MNI152 midline into subject space using the MNI152-to-subject deformation field (Section 4.3).

3. Anatomical segmentations (Section 4.2) are used to extract VASARI (Visually AcceSAble Rembrandt Images) features using a modified VASARI-auto pipeline (Ruffle et al., 2024), yielding standardized tumor- and anatomy-specific descriptors (Section 4.4).

4. A general-purpose LLM is provided a context-specific prompt and tasked with generating structured radiology reports, with generation conditioned on deterministic BTReport-extracted features (Section 4.6).

## 4.2. Segmentation Statistics

**BTReport Inputs:** For a given case, inputs to the BTReport framework include their T1 sequence and corresponding tumor segmentation masks. Tumor segmentation masks are derived from manual annotation or automated segmentation, and tumor sub-region annotations (necrotic core, edema, enhancing tumor) follow BraTS convention.

**Robust Anatomical Segmentations & Volumetric Statistics:** Anatomical segmentations provide tumor-ROI relational context by localizing tumors with respect to critical brain structures. However, many automated methods assume near-normal anatomy and can fail when large tumors deform tissue and obscure boundaries. Prior work improves tumor-robust segmentations either by training with synthetic tumor-mask augmentation (Lei et al., 2024), or by synthesizing pseudo-healthy structural images that suppress tumor appearance, increasing the reliability of established segmentation pipelines (Iglesias et al., 2023). Our approach closely resembles the latter.

First, we register the MNI152 atlas (Collins et al., 1999; Fonov et al., 2011, 2009) to subject space using SynthMorph (Hoffmann et al., 2024). The resulting warped atlas, referred to as MNI152-to-subject, provides a pseudo-healthy anatomical representation of the brain. Next, we obtain anatomical segmentations by running SynthSeg (Billot et al., 2023) on the MNI152-to-subject volume. As SynthSeg operates on the tumor-free brain representations, anatomical segmentations remain robust when tumors deform native anatomy. Finally, anatomical labels are merged with the tumor and midline segmentations, to produce a unified mask that jointly represents normal neuroanatomy and pathological structures. With this joint segmentation, we can reliably extract quantitative features including tumor and ventricle volume, lesion count, tumor sub-region proportions, and tumor-ROI overlap.

## 4.3. 3D Midline Shift Measurement via Atlas-based Segmentation

Midline shift (MLS) is an intracranial pathology characterized by the displacement of brain tissue across the skull's midsagittal axis. MLS arises as a result of traumatic brain injury or tumor mass effects and is an indirect indicator of elevated intracerebral pressure. Estimation of MLS is done by identifying the axial slice with the largest deviation, as indicated by midline structures such as the septum pellucidum, the third and fourth ventricles, and the falx cerebri. However, this estimation is subject to high inter-rater variability as there is not a standard procedure for axial slice level selection.

Here, we propose a novel pipeline for MLS estimation based on clinical guidelines, using a deep learning atlas-based segmentation approach. Our approach, shown in Figure 2, leverages the robust registration capabilities of SynthMorph (Hoffmann et al., 2024) to register hand-annotated midline segmentations from a MNI152 atlas template onto patient

T1 scans. These are compared to an "ideal" midline defined by connecting the anterior and posterior points of the falx cerebri for each axial slice. By calculating the distance between the ideal and subject midlines at each voxel, we obtain accurate 3D MLS estimations in seconds, giving a more complete picture in comparison to 2D automated or manual methods. This approach has strong zero-shot generalization and can be applied to MRI or CT scans.

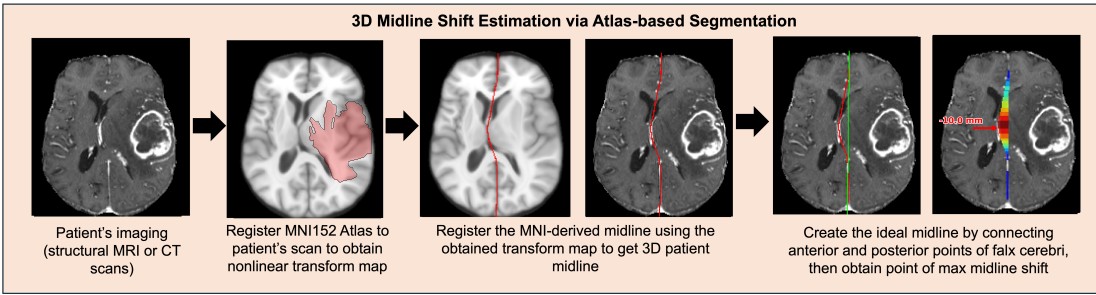

Figure 2: Atlas-based 3D midline shift (MLS) estimation using SynthMorph (Hoffmann et al., 2024), in which atlas midline annotations are registered to patient imaging and voxel-wise distances to an ideal midline axis are computed per axial slice.

### 4.4. VASARI Feature Extraction

To standardize neuroimaging-derived glioma characterization and improve repeatability, the VASARI (Visually AcceSAble Rembrandt Images) feature set was developed by The Cancer Imaging Archive. The VASARI feature set uses controlled vocabulary to quantitatively describes anatomical relationships between GBM tumors and clinically relevant brain structures established in the literature. Furthermore, these relationships are also routinely included in neuroradiology reports and used by neurosurgeons to assess whether patients are candidates for surgical intervention. VASARI features have been used to accurately predict tumor histological grade (WHO I-IV grade), disease progression, molecular mutation status (e.g., *IDH1* WT/mutant, *MGMT* un/methylated), risk of recurrence, and overall patient survival (Jain et al., 2014; Nicolasjilwan et al., 2015; Peeken et al., 2019; Setyawan et al., 2024; Wang et al., 2021; Zhou et al., 2017). Here, we employ a modified variant of VASARI-auto (Ruffle et al., 2024), an automated labeling tool, which has been validated as non-inferior to radiologist VASARI annotations, and can be used to reduce inter-rater variance. The included variations use the subject-space anatomical and midline segmentations extracted in Sections 4.2 and 4.3 to improve feature accuracy.

### 4.5. BTReport Feature Validation

To determine whether extracted imaging features are clinically relevant, we assessed them based on two criteria: (1) whether features correspond to commonly reported radiological concepts (Section 4.5.1) and (2) by modeling their prognostic value with respect to patient survival (Section 4.5.2). To validate the accuracy of extracted features, we assess the quality of feature extraction in generated reports to those found in associated radiologist-authored ground truth reports, assessing categorical and numerical feature precision and error.

### 4.5.1. Semantic Clustering of Common Radiology Findings

Here, we derived a ranked list of the most frequently included topics in real radiology reports. First, we extract all discrete factual claims across all reports in the HuskyBrain dataset using the method proposed in TBFact (Blondeel et al., 2025). TBFact uses an LLM (in this case DeepSeek-R1 (Guo et al., 2025)) to divide each clinical report into independently verifiable factual claims. Next, we embedded each extracted claim using a lightweight sentence-transformer model (all-MiniLM-L6-v2 (Reimers and Gurevych, 2019)) to obtain dense semantic representations. All embedded claims were pooled across the dataset and clustered using hierarchical agglomerative clustering (cosine distance, average linkage) without specifying the number of clusters a priori, allowing groups of semantically related statements to emerge from the data in an unsupervised manner.

We then use a pre-trained Gemma 3 27B LLM (Gemma Team et al., 2025) model to summarize claims in each cluster into representative topic sentences. For example, a cluster containing claims such as: ["The lesion measures 4.0 x 3.5 cm", "The lesion measures 3.9 x 2.0 x 2.1 cm.", ...], is summarized by the topic sentence "Lesion size and measurements reported." The number of claims assigned to each cluster serves as an estimate of the frequency of the corresponding topic. The 35 most frequent cluster descriptors, along with representative example claims for each cluster, are reported in Appendix A.

Collectively, the resulting set of cluster descriptors and frequencies provides an interpretable, data-driven view of radiological concepts based on real-world reporting. We use these concepts to validate the clinical relevance of deterministically extracted features found in BTReport and assess their alignment with findings in real radiology reports.

### 4.5.2. Survival Outcome Modeling

To assess whether BTReport-extracted features were predictive of clinical outcomes, we evaluated their association with overall survival measured from diagnosis. This analysis used the dataset described in Section 3.2, which provides survival outcomes and patient metadata for a subset of BraTS cases. Survival analyses were performed using Kaplan–Meier estimators implemented with the `lifelines` library (Davidson-Pilon, 2024), with curves reported in Appendix B. Differences between survival groups were assessed using the log-rank test, and relative risk was quantified using Cox proportional hazards models, reporting hazard ratios with 95% confidence intervals. Statistical significance was determined from p-values derived from both the log-rank tests and Cox models.

### 4.5.3. Clinically Relevant Features Included in BTReport

Table 1 summarizes the quantitative imaging features used by BTReport for automated report generation. Each feature was evaluated according to two clinically motivated criteria: 1) whether it corresponded to at least one of the top 35 radiological concepts most frequently documented in reference reports from the HuskyBrain dataset (Section 4.5.1), and 2) whether it was a statistically significant predictor of patient survival time based on the survival analysis described in Section 4.5.2. We found that 21/22 of the BTReport features are commonly reported in clinical reports, and 11/22 are predictive of overall patient survival. This analysis highlights the clinical relevance of extracted features, motivating their use as inputs for downstream RRG.

Table 1: Summary of quantitative features used in BTReport. Feature groups are color-coded: gray = segmentation statistics, green = VASARI features, orange = midline features. Key: ● indicates the feature is among the top 35 most frequently reported concepts in real radiology reports, and ● indicates the feature is a significant survival predictor. Acronyms: WM-white matter; MLS-midline shift.

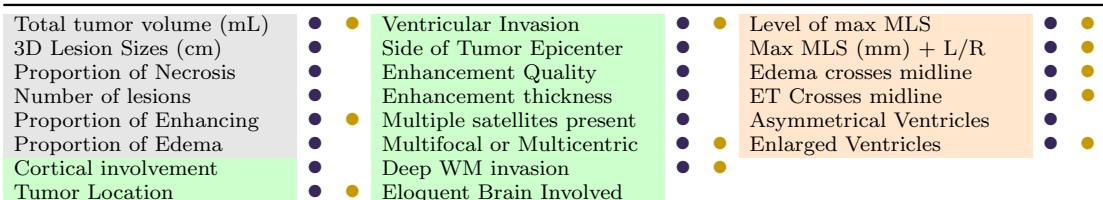

## 4.6. Structured Synthetic Report Generation Using LLMs

The BTReport framework generates the *Findings* sections of radiology reports in a style that mimics a target institution through an in-context learning prompt to provide stylistic guidance and emphasize the relevant radiological facts which should be included in generated reports (Appendix C). Some facts in reference reports (e.g., diffusion characteristics, vascular features) cannot be derived from standard structural mpMRI modalities used for tumor segmentation (T1w, T1c, T2w, T2-FLAIR). Therefore, the LLM is explicitly instructed to exclude these findings in generated reports to prevent unsupported claims.

Report generation is conditioned on the BTReport feature set and prompt instructions that restrict the LLM to narrative synthesis and stylistic alignment, enabling tighter control over report content and structure without task-specific fine-tuning. This prompting strategy improves interpretability by allowing reported findings to be directly traced to deterministic features.

For RRG with BTReport, we experiment with two open-source pre-trained LLMs with reasoning capability: gpt-oss:120b (Agarwal et al., 2025) and Llama 3.1 70B Instruct (Grattafiori et al., 2024). These models were selected based on their strong reasoning and instruction-following capabilities. To avoid cloud sharing of medical data, only local offline models were used. Examples *Findings* sections from synthetically generated reports across the different BTReport variants and other RRG frameworks can be found in Appendix F.

## 5. Evaluation

We evaluated BTReport on 129 paired GBM image-report cases from the HuskyBrain dataset. Synthetic reports generated by BTReport and other neuro-oncology RRG frameworks described in Section 2 were compared against ground-truth radiology reports using automated metrics (Section 5.1), and the reliability of BTReport-extracted features was assessed (Section 5.2). To test robustness to segmentation quality, we compared BTReport outputs generated using DeepMedic (Kamnitsas et al., 2017) segmentations, a dual-pathway CNN-based model, with outputs generated using segmentations from the BraTS 2023-winning submission (Ferreira et al., 2024), which is based on an nnU-Net (Isensee et al.,

2021) ensemble. Prior work (Pemberton et al., 2023) has shown nnU-Net–based models to outperform DeepMedic for brain tumor segmentation, motivating this comparison.

### 5.1. Automated Evaluation of Generated Radiology Reports

Generated radiology reports were evaluated using RadEval (Xu et al., 2025), a unified open-source framework that evaluates radiology text based on:

- N-gram-based lexical similarity: BLEU (Papineni et al., 2002), ROUGE (Lin, 2004)
- pre-trained contextualized embeddings: BERTScore (Zhang et al., 2020)
- clinically grounded scores: RaTEScore (Zhao et al., 2024)

To assess the clinical correctness of content in generated reports, we used TBFact (Blondeel et al., 2025), an LLM-based factuality metric that evaluates reports based on factual inclusion, distortion, and omission. These metrics are presented in Tables 3 and 4.

### 5.2. Reliability of Extracted Features

We evaluated the reliability of BTReport-extracted clinical features by comparing them with the same features extracted from ground-truth radiology reports in the HuskyBrain dataset. Using LangExtract (Goel, 2025), we decompose radiology reports into class-attribute pairs, where each class represents a clinically defined radiology concept, and their associated attribute(s) specify the characteristic linked to the concept. For example, the text ["arising in the right parietal lobe..."] is decomposed into the "side of tumor epicenter" class and the "right" attribute (See Appendix E for a complete example). We present the accuracy of categorical features and the error of the numerical features in Table 2.

## 6. Results

### 6.1. Quality of Extracted Features

Table 2: Attribute-level reliability of BTReport-extracted features relative to ground-truth HuskyBrain radiology reports. Categorical features report accuracy, and numeric features report mean absolute error (MAE) ± standard deviation. Best-performing variant is highlighted in blue.

| Method | Accuracy | | | MAE | | |
|---|---|---|---|---|---|---|
| | Cortical involvement | Side of tumor epicenter | Ventricular effacement | Midline shift (mm) | Number of lesions | Tumor volume (cm$^3$) |
| BTReport (DM) | 0.79 | **1.00** | **0.56** | 1.26 ± 2.60 | 1.48 ± 2.63 | **1.80** ± 3.95 |
| BTReport (FI) | **0.80** | **1.00** | 0.52 | **1.21** ± 2.55 | **0.41** ± 1.69 | 2.00 ± 5.16 |

DM: DeepMedic (Kamnitsas et al., 2017) segmentations    FI: Faking It (Ferreira et al., 2024) segmentations

Table 2 reports the reliability of BTReport-extracted clinical features under two segmentation inputs. Across both variants, categorical features showed strong agreement with LangExtract-derived ground truth, including perfect accuracy for identifying the side of the tumor epicenter and high accuracy for cortical involvement and tumor location. Ventricular effacement was less reliable, indicating that this attribute may be more challenging to

describe consistently. For numeric features, BTReport achieved low mean absolute error for lesion count and midline shift, with consistently lower errors when using the improved segmentation model (FI). Tumor volume errors were comparable across variants.

## 6.2. Lexical Similarity of Generated Reports

Table 3: Mean ± standard deviation BLEU and ROUGE metrics. Best values per metric are highlighted in blue. [†]Following Approximate Randomization, synthetic reports generated with all BTReport variants were superior to those generated using other frameworks across all evaluation metrics ($p < 0.0001$).

| Framework | BLEU-1 | BLEU-2 | ROUGE-1 | ROUGE-2 |
|---|---|---|---|---|
| BTReport (gpt-oss:120B, DM)[†] | 0.236 ± 0.068 | 0.124 ± 0.044 | 0.360 ± 0.078 | 0.109 ± 0.038 |
| BTReport (gpt-oss:120B, FI)[†] | 0.244 ± 0.070 | 0.132 ± 0.044 | **0.371** ± 0.078 | **0.115** ± 0.040 |
| BTReport (LLaMA3:70B, DM)[†] | **0.248** ± 0.078 | **0.136** ± 0.052 | 0.362 ± 0.080 | 0.115 ± 0.043 |
| AutoRG-Brain (Lei et al., 2024) | 0.158 ± 0.072 | 0.080 ± 0.042 | 0.268 ± 0.060 | 0.070 ± 0.032 |
| Seg-to-Exp (Valerio et al., 2025) | 0.085 ± 0.039 | 0.035 ± 0.018 | 0.163 ± 0.055 | 0.023 ± 0.013 |

DM: DeepMedic (Kamnitsas et al., 2017) segmentations    FI: Faking It (Ferreira et al., 2024) segmentations

Table 3 summarizes the lexical similarity between generated and reference reports using mean BLEU and ROUGE metrics calculated over the 129 subject test dataset. Across all metrics, all BTReport variants substantially outperformed *AutoRG-Brain* and *Seg-to-Exp*, indicating closer n-gram overlap with clinical ground-truth reports. Across the two BTReport variants, BTReport with LLaMA3:70B achieved the highest BLEU-1 and BLEU-2 scores, indicating improved short-range lexical precision. In contrast, BTReport with gpt-oss:120B attained the highest ROUGE-1 and ROUGE-2 scores, suggesting improved recall of clinically relevant phrases and longer contextual spans. While the relative strengths marginally differed across metrics for BTReport variants, all demonstrated statistically significant improvements in lexical alignment with reference reports in comparison to baseline methods.

## 6.3. Factual Accuracy of Generated Reports

Table 4 reports factual consistency and semantic alignment of generated reports using TB-Fact, BERTScore, and RaTEScore. Across all metrics, both BTReport variants outperformed *AutoRG-Brain* and *Seg-to-Exp*, indicating improved factual grounding and clinically coherent report generation. BTReport with gpt-oss:120B and LLaMA3:70B achieved the highest overall performance, with the gpt-oss:120B model showing consistently higher TBFact, BERTScore, and RaTEScore values under the higher-quality segmentation input (FI) compared to DM. This suggests that reports generated with BTReport contain a greater proportion of verifiable clinical statements that are consistent with the underlying imaging findings. Both BTReport frameworks achieved substantially higher scores for the BERTScore and RaTEScore metrics, indicating closer semantic alignment with reference reports and improved clinical relevance. In contrast, *AutoRG-Brain* exhibited moderate

performance, while *Seg-to-Exp* showed limited factual consistency, reflected by low TBFact scores and reduced semantic similarity. All improvements using the BTReport framework were statistically significant ($p < 0.0001$) when evaluated with AR.

Table 4: Mean $\pm$ standard deviation TBFact, BERTScore, and RaTEScore metrics. Best values per metric are highlighted in blue. [†] Following Approximate Randomization, synthetic reports generated with all BTReport variants were superior to those generated using other frameworks across all evaluation metrics ($p < 0.0001$).

| Framework | TBFact (DeepSeek-R1) | | | | BERTScore | RaTEScore |
|---|---|---|---|---|---|---|
| | Score | Prec. | Recall | F1 | | |
| BTReport (gpt-oss:120B, DM)[†] | $0.313 \pm 0.145$ | $0.345 \pm 0.155$ | $0.325 \pm 0.163$ | $0.313 \pm 0.145$ | $0.447 \pm 0.060$ | $0.568 \pm 0.057$ |
| BTReport (gpt-oss:120B, FI)[†] | $\mathbf{0.353} \pm 0.151$ | $\mathbf{0.412} \pm 0.146$ | $\mathbf{0.349} \pm 0.162$ | $\mathbf{0.359} \pm 0.145$ | $\mathbf{0.453} \pm 0.055$ | $\mathbf{0.577} \pm 0.054$ |
| BTReport (LLaMA3:70B, DM)[†] | $0.295 \pm 0.130$ | $0.377 \pm 0.168$ | $0.274 \pm 0.136$ | $0.295 \pm 0.130$ | $0.433 \pm 0.062$ | $0.567 \pm 0.061$ |
| AutoRG-Brain (Lei et al., 2024) | $0.072 \pm 0.123$ | $0.282 \pm 0.155$ | $0.186 \pm 0.137$ | $0.196 \pm 0.130$ | $0.327 \pm 0.047$ | $0.477 \pm 0.053$ |
| Seg-to-Exp (Valerio et al., 2025) | $0.014 \pm 0.047$ | $0.131 \pm 0.121$ | $0.147 \pm 0.108$ | $0.098 \pm 0.089$ | $0.156 \pm 0.042$ | $0.409 \pm 0.039$ |

DM: DeepMedic (Kamnitsas et al., 2017) segmentations    FI: Faking It (Ferreira et al., 2024) segmentations

## 7. Discussion

We present BTReport, a framework for brain tumor RRG grounded in clinically relevant quantitative imaging features. Overall, our findings support the two-stage report generation paradigm for neuro-oncology RRG, suggesting that in medical imaging domains with limited data, adding quantitative features to prompts is an efficient way to generate reports and improve factual consistency. BTReport enables accurate measurement-grounded reporting without requiring task-specific fine-tuning of vision encoders or VLMs, and generates interpretable reports by using modular, reliably-extracted features.

We presented novel pipelines for midline shift measurement and tumor-robust anatomical segmentation for deterministic feature extraction, and showed that these features had strong agreement with expert annotation. Additionally, other quantitative features were selected based on clinical guidelines such as VASARI and validated to be significant predictors of survival time using KM-analysis. By clustering concepts frequently reported in real radiology reports, we validated that the included features were clinically relevant. When compared with existing neuro-oncology RRG approaches, BTReport generated reports are superior in terms of lexical similarity and factual accuracy. We believe this improvement is due to the clinically-relevant features used for generation and the in-context learning prompt which allows reasoning LLMs to predict the most relevant features for RRG.

To facilitate further research in neuro-oncology RRG, we provide BTReport-BraTS, an open-source companion dataset containing anatomical descriptors, metadata, and BTReport generated reports for mpMRI cases in the BraTS'23 dataset. To assess the clinical applicability of BTReport, we have developed BTReview, a survey tool for radiologist assessment of generated neuro-oncology radiology reports (Appendix H). Future work will obtain radiologist feedback with BTReview, incorporate additional features such as white matter hyperintensities and basal cistern status, handle additional mpMRI imaging modalities, and include descriptions of ischemic or hemorrhagic stroke findings.

## Acknowledgments

The authors are grateful to Caitlin Neher, Leonardo Schettini, and Ankush Jindal for their helpful discussions. The authors are grateful for Dr. Kambiz Nael for his clinical perspectives for improving the evaluation platform. The authors are also grateful to James K. Ruffle, for making the VASARI-Auto implementation available.

## 8. Ethics

This study's activities were approved by the Institutional Review Board at the University of Washington (STUDY00022466). This research is in accordance with the principles embodied in the Declaration of Helsinki.

## 9. Funding Statement

The work of Juampablo Heras Rivera was partially supported by the U.S. Department of Energy Computational Science Graduate Fellowship under Award Number DE-SC0024386.

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

# Appendix A. Cluster Analysis of Common Topics in Reference Reports

| Cluster Description | Frequency | Two Example Sentences in this cluster |
|---|---|---|
| Lateral ventricle effacement/asymmetry; possible... | 50 | "The lateral ventricles are symmetric.", "The lateral ventricles are symmetric" |
| Midline shift presence and magnitude.... | 27 | "There is a 2 mm left midline shift", "There is a susceptibility millimeters rightward midline shift." |
| Cisterns appear normal, no obstruction. | 19 | "The basal cisterns are patent.", "The basal cisterns are patent" |
| Peritumoral/Vasogenic edema present. *(This... | 18 | "There is mild-to-moderate associated edema", "The first mass has associated surrounding vasogenic edema" |
| Lesion size and measurements reported. | 16 | "The lesion measures 4.0 x 3.5 cm", "The lesion measures 3.9 x 2.0 x 2.1 cm." |
| No acute intracranial hemorrhage or infarct. | 15 | "There is no acute infarct", "No acute infarct is seen" |
| Multiple intracranial masses present bilaterally.... | 15 | "There is a large irregular enhancing mass centered in the right frontal lobe", "There is a right frontal lobe mass" |
| Multiple, enhancing intracranial lesions present. | 14 | "The lesion originates in the anterior paramedian left frontal lobe", "There is a lesion in the right frontal lobe" |
| White matter disease, likely nonspecific etiology. | 14 | "There are scattered deep and periventricular white matter T2/FLAIR hyperintensities", "There is mild subcortical and periventricular white matter T2 FLAIR abnormality" |
| No mass effect/midline shift. (Alternatively: No... | 13 | "There is no shift in the brain", "There is no shift of the brain structures" |
| Ventricular system: normal or prominent.... | 13 | "The ventricles, sulci, and cisterns are normal", "The remaining ventricles, sulci, and cisterns are normal" |
| Restricted diffusion within the lesion(s). | 9 | "The solid components of the lesion demonstrate moderate diffusion restriction", "The lesion has peripheral areas of mild diffusion restriction" |
| Frontal/Temporal lobe FLAIR signal abnormality | 9 | "There is extensive T2/FLAIR signal abnormality in the right frontotemporal lobes", "There is surrounding FLAIR signal hyperintensity inferiorly going into the temporal lobe and posteriorly." |
| Herniation syndromes present on imaging. (or... | 8 | "There is suggestion of transtentorial herniation", "There is leftward subfalcine herniation" |
| Mass dimensions and size measurements. | 8 | "The mass measures 2.4 x 3.4 cm in axial cross-section", "The mass measures approximately 6.6 x 4.7 cm in transverse dimensions and 4.6 cm craniocaudally" |
| Uncal medialization, potentially impacting... | 8 | "There is right uncal medialization", "There is medialization of the right uncus" |
| No acute intracranial hemorrhage present.... | 7 | "No parenchymal hemorrhage is present", "There is no associated hemorrhage" |
| Mass effect on lateral ventricles. | 7 | "The mass extends along the ependymal surface of the right lateral ventricle", "The mass extends into the posterior horn of the right lateral ventricle" |
| Edema predominantly affecting frontal & temporal... | 6 | "There is perilesional edema along the predominantly anterior aspect of the medial frontal lobes", "The vasogenic edema extends to the frontal lobe" |
| Mass size and measurements. (Alternatively: Lesion... | 6 | "The mass measures 40 x 53 mm", "The mass measures approximately 58 x 44 x 44 mm" |
| Corpus callosum lesion, midline crossing/spread. | 6 | "The lesion extends into the splenium of the corpus callosum, crossing midline to the right", "There is ependymal spread along the body of the corpus callosum" |
| Diffusion restriction presence/absence &... | 6 | "There is no associated restricted diffusion", "There are areas of internal diffusion restriction and susceptibility" |
| Frontal horn effacement & ventricular asymmetry. | 6 | "There is partial effacement of the right frontal horn", "There are areas of subtle ependymal enhancement in the bilateral frontal horns" |
| Hemorrhagic lesion with restricted diffusion. | 5 | "The lesion restricts diffusion and has intralesional hemorrhage", "The mass is T2 hyperintense, contains multiple foci of internal hemorrhage, and demonstrates mottled diffusion restriction consistent with hypercellularity and/or necrosis." |
| Cistern effacement suggests mass effect. (Or, more... | 5 | "There is effacement of the right crural cistern", "There is partial effacement of the basal cisterns" |
| Sulcal effacement, widespread cortical... | 5 | "There is sulcal effacement involving the right parietal, posterior temporal, and occipital lobes", "There is mild sulcal effacement of the left occipital lobe." |
| Mass shows restricted diffusion on imaging. (Or,... | 5 | "There are patchy foci of restricted diffusion within the mass", "The second mass has diffusion restriction" |
| Corpus callosum mass/involvement. (or simply:... | 5 | "The mass extends into the right-sided genu of the corpus callosum", "The mass extends along the splenium of the corpus callosum" |
| Basal ganglia involvement with signal abnormality. | 5 | "The signal abnormality extends into the right basal ganglia, right thalamus, right cerebral peduncle, and right midbrain", "The hyperintensity involves the bilateral basal ganglia, with greater involvement on the left" |
| Ventricular size and morphology assessment. | 4 | "The third ventricle is near slitlike", "There is complete effacement of the third ventricle" |
| Midline shift at foramen of Monro | 4 | "There is a negative millimeters leftward midline shift at the level of the foramen of Monroe", "There is some millimeters of rightward midline shift at the level of the foramen of Monro" |
| No acute hydrocephalus present. | 4 | "There is no evidence of acute hydrocephalus", "There is no hydrocephalus" |
| Corpus callosum FLAIR edema/hyperintensity... | 4 | "The surrounding T2/FLAIR signal is similar and extends to the left splenium, septum pellucidum, and superior corpus callosum", "There is subtle patchy T2 FLAIR hyperintensity along the right body of the corpus callosum" |
| Peripheral mass enhancement characteristics. (Or... | 4 | "The first mass has peripheral enhancement with a nodular solid enhancing component", "The mass has irregular somewhat nodular peripheral enhancement" |
| Right cerebral peduncle lesion/mass effect. | 4 | "There is mass effect on the right cerebral peduncle", "The lesion has questionable extension into the posterior right cerebral peduncle" |

Table 5: Example sentences associated with the top 35 radiological concept clusters ranked by their prevalence in reference radiology reports.

## Appendix B. Kaplan-Meier Survival Analysis of Extracted Features

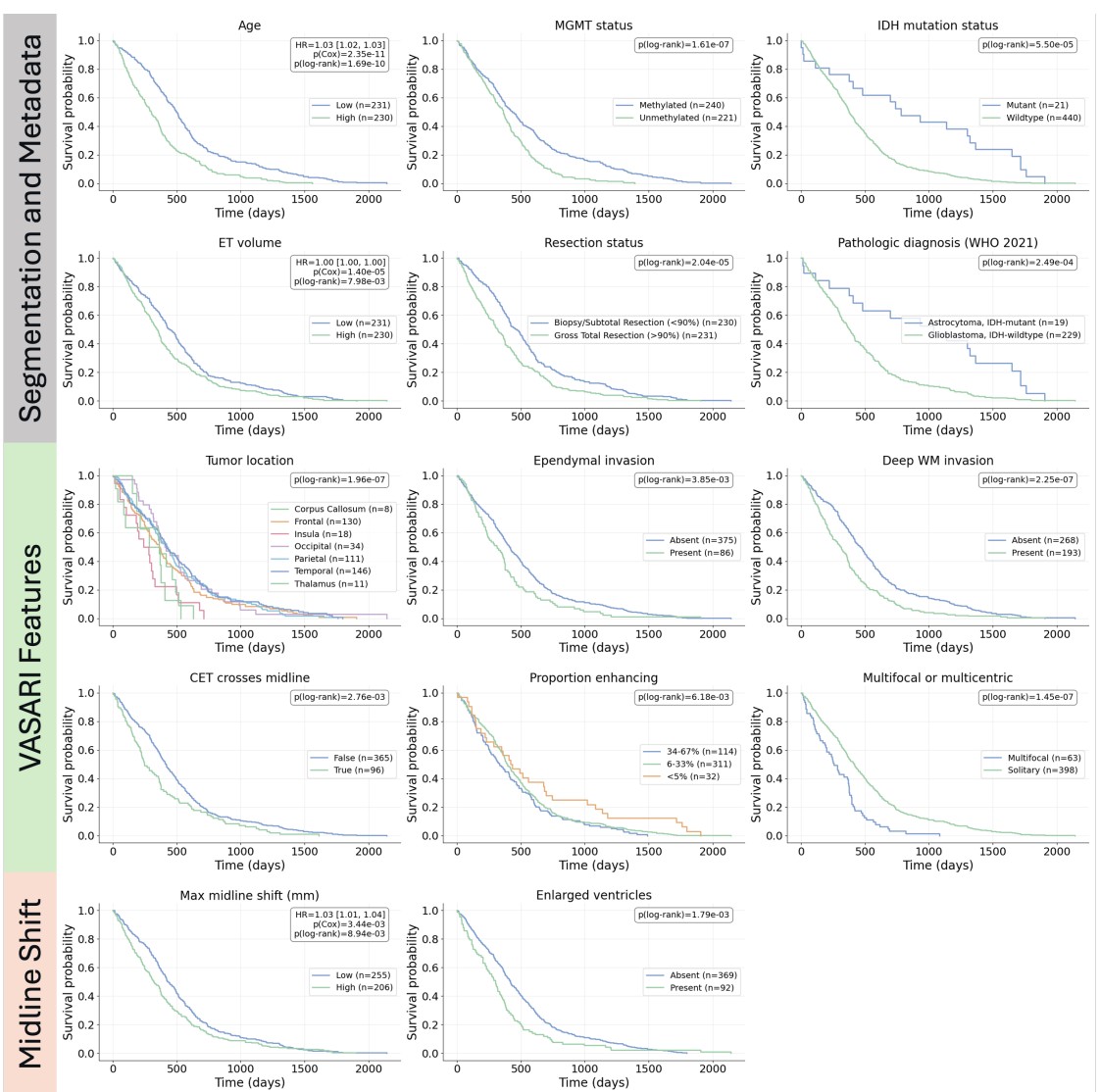

Figure 3: Kaplan-Meier (KM) plots for demographic and radiogenomic features, and deterministic features extracted with the BTReport framework. Survival probabilities at specific time points are obtained by projecting vertically from the time of interest to the curve and horizontally to the y-axis. Example: for the Age feature, at 500 days post-diagnosis, the estimated survival probability for the older age group (High) is ∼ 20%, compared to ∼ 50% for the younger age group (Low). Kaplan–Meier analyses indicate that many of these features are predictive of overall survival, highlighting their clinical relevance and motivating their use as structured inputs for radiology report generation.

## Appendix C. LLM Prompt for Radiology Report Generation

---

**BTReport Prompt**

---

You are a radiologist generating a synthetic clinical MRI report.
Below are example FINDINGS sections taken from real brain tumor reports:

**EXAMPLE FINDINGS:**
{example_findings}

---

Your job is to generate a FINDINGS section in the same clinical style, but only using the METADATA provided below.
Please abide strictly to the following rules (follow them exactly).

1. Use only the metadata provided for quantitative statements. Do NOT hallucinate any information that is not directly inferable.

2. Include 10–20 clinically meaningful findings summarized in an anatomically descriptive manner. Prioritize describing abnormal or clinically significant observations.

3. Preserve the subsection structure from the example reports. Make sure to include the following subsections: MASS EFFECT & VENTRICLES and BRAIN / ENHANCEMENT.

4. Never mention imaging sequences other than T1n, T2w, T2 FLAIR, or T1-Gd. Do not mention diffusion, perfusion, spectroscopy, MRA, or other modalities (unless stated explicitly in metadata).

5. Do not mention structures or measurements not present in the metadata.

6. Mandatory Considerations: Make sure to include the following findings below if present in metadata. Remember to follow the sentence structure in the example reports.

    a) Maximum midline shift represented in mm units.

        i) Make sure to describe the magnitude and the direction of the midline shift.

        ii) Describe the anatomical level at which the (e.g., foramen of Monro, third and fourth ventricles, septum pellucidum).

        iii) If shift is minimal (e.g., $< 5$ mm), explicitly state the measurement as no shift, but still provide the measurement.

    b) If tumor mass effect is present, describe the mass effect on ventricles or surrounding brain structures.

        i) Include a description of ventricular effacement (if present), including which horn (anterior/posterior horn), and on which hemisphere it is observed.

    c) Comment on ventricular status. If effacement is present, describe the extent of the effacement or asymmetry. If ventricles are normal, explicitly state so (mirroring example reports). Use the following metadata fields in your description: "Asymmetrical Ventricles", "Enlarged Ventricles".

    d) Describe the size of the primary lesion, as well as any smaller secondary lesions (if present) represented in cm units. Use the 3D measurements from the metadata. Make sure to include the following:

        i) If multiple lesions exist, summarize number, dominant lesion, and laterality. Use the following metadata fields in your description: "Number of lesions" and "Multifocal or multicentric."

        ii) Anatomical location of lesion(s).

        iii) Use the following metadata fields in your description: "Tumor Location", "Side of Tumor Epicenter", and "Region Proportions."

    e) Describe enhancing characteristics. Use the following metadata fields in your description: "Enhancement quality", "Thickness of enhancing margin", "Proportion Enhancing".

        i) Describe enhancement style (e.g., rim-enhancing, mildly enhancing, peripheral enhancing, multilobular enhancing) only if explicitly supported.

        ii) Describe edematous tissue. Use the following metadata fields in your description: "ED volume", Whether edema crosses midline, "Proportion of edema".

        iii) Describe vasogenic edema and its extent only if metadata supports it.

    f) Describe invasion and involvement. Use the following metadata fields in your description: "Cortical involvement", "Deep WM invasion", "Ependymal invasion", "Eloquent Brain Involvement".

    g) Describe the necrosis if present. Use the metadata field "Proportion Necrosis" to describe the central foci of necrosis.

**METADATA (for subject {subject_id}):**
{metadata_json}

---

Write the **FINDINGS** section now, using clinical radiology language.

## Appendix D. Short version of LLM Prompt for BTReport

---

**BTReport Prompt (Short)**

You are a radiologist generating a synthetic clinical MRI report.
Below are example FINDINGS sections taken from real brain tumor reports:

**EXAMPLE FINDINGS:**
{example_findings}

---

Now generate a similar FINDINGS section, but **only** using the metadata provided below.
- Do not hallucinate any information that is not directly inferable.

- Preserve all subsections present in the example findings reports.

- Select the top 10 metadata-supported findings; real reports typically include 7–10 facts.

- Prioritize abnormal or clinically significant findings.

- Only T1n, T2w, T2 FLAIR, and T1-Gd sequences were obtained; do not comment on diffusion or other modalities.

- Comment on midline shift in the style of the example reports, including direction and magnitude.

- Describe mass effect and ventricular effacement if present, including laterality and anterior/posterior horn involvement.

- Report lesion dimensions in three orthogonal axes (cm) if supported by metadata.

- Do not mention structures, measurements, or features unless supported by the metadata.

**METADATA (for subject {subject_id}):**
{metadata_json}

---

Write the **FINDINGS** section now, using clinical radiology language.

---

## Appendix E. Radiology report feature extraction for validating BTReport features

Figure 4: Visualization of features extracted from free text radiologist-authored report using LangExtract.

Figure 5: Visualization of features extracted from free text BTReport-generated report using LangExtract.

## Appendix F. Example Generated Reports

### Model comparison of RRG output: Findings Section

**COLOR KEY:** Green indicates correct statements; yellow partially correct; red incorrect; gray ground-truth statements referenced by generated reports; ** findings derived from unavailable MRI modalities.

#### Ground Truth Findings

**MASS EFFECT & VENTRICLES:** Prominent leftward midline shift by approximately 14 mm (XXX/XXX). There is also medialization of right uncus. The basal cisterns are partially effaced.

**BRAIN/ENHANCEMENT:** A large irregular enhancing lesion centered within the right temporal lobe with significant mass effect. Restricted diffusion** is noted within the enhancing portion of the lesion, which contains susceptibility artifact suggestive of microhemorrhages/angioinvasion.** The lesion measures approximately 6.8 x 4.4 x 4.8 cm (AP, TV, CC). There is probable small subependymal enhancement at the atria of right lateral ventricle (XXX/XXX, XXX/XXX). No acute hematoma or infarct is seen.

#### Generated Findings

| Framework | Prediction |
|---|---|
| **BTReport** (gpt-oss:120B) | ROUGE: 0.37  BLEU: 0.24  RATESCR: 0.56  BERTSCR: 0.47  TB-F1: 0.33 
 **MASS EFFECT & VENTRICLES:** Approximately 12 mm of right-to-left midline shift is present, measured at the level of the falx cerebri above. The ventricles are asymmetrical, with compression of the right lateral ventricle secondary to tumor-related ependymal invasion; there is no overall ventricular enlargement. No tonsillar herniation is identified. 
 **BRAIN/ENHANCEMENT:** A solitary right-sided lesion involving the temporal, cortical, and parietal lobes measures 7.1 x 5.6 x 5.3 cm (AP x TV x CC). The mass shows marked heterogeneous enhancement with a thick ($> 3mm$) enhancing margin; roughly 28% of the tumor volume enhances. A necrotic core comprises about 22% of the lesion volume. Deep white-matter invasion and cortical involvement are evident, and there is direct ependymal invasion of the adjacent lateral ventricle. The enhancing component does not cross the midline, but extensive vasogenic edema (84 mL, around 50% of total lesion volume) crosses the midline and surrounds the lesion, extending into the contralateral hemisphere. Multiple small enhancing satellite nodules are present along the cortical surface. The edema involves visual-association cortex, indicating eloquent brain (vision) involvement. |

**BTReport**
**(LLaMa3:70B)**

ROUGE: 0.42  BLEU: 0.29  RATESCR: 0.57  BERTSCR: 0.47  TB-F1: 0.34

**MASS EFFECT & VENTRICLES:** There is a maximum of 12 mm leftward midline shift at the level of the falx cerebri above. The ventricles are asymmetrical, with tumor-side ventricular asymmetry noted. There is no enlargement of the ventricles.

**BRAIN/ENHANCEMENT:** Within the right temporal, cortex, and parietal lobes, there is a solitary, predominantly enhancing mass measuring 7.1 x 5.6 x 5.3 cm. The lesion exhibits marked enhancement with a thick ($> 3mm$) enhancing margin. There are multiple small satellite lesions present. The enhancing region does not cross the midline. The lesion invades the cortex and deep white matter, with ependymal invasion also present. There is a large volume of surrounding vasogenic edema (83.6 mL), which crosses the midline. The proportion of edema is 49.74%. A central focus of necrosis is present, comprising approximately 21.99% of the lesion."

**AutoRG-Brain**

ROUGE: 0.32  BLEU: 0.25  RATESCR: 0.53  BERTSCR: 0.41  TB-F1: 0.25

After contrast administration, the lesion shows significant enhancement on T1-weighted images with unclear boundaries. The adjacent brain parenchyma shows significant edema in the brain pools and ventricles. After contrast administration, the lesion in the right temporal lobe demonstrates significant ring-like enhancement. The lesion measures approximately 82*111*82mm, with indistinct borders and a large amount of edema in the adjacent brain tissue, causing compression of the right lateral ventricle and deviation of midline structures to the left.

**Seg-to-Exp**

ROUGE: 0.21  BLEU: 0.16  RATESCR: 0.41  BERTSCR: 0.26  TB-F1: 0.0

The brain tumor segmentation results indicate that the tumor is primarily located in the right temporal and parietal regions, with a significant presence in the Temporal-to-Parietal (GapMap) region (40.03% of the tumor). The tumor also affects several surrounding regions, including Area STS2 (STS), Area STS1 (STS), Frontal-to-Temporal-II (GapMap), and Area TE 2.2 (STG). Semantic segmentation of the tumor reveals three distinct categories: **Tumor Core** (red), **Peritumoral Edema** (yellow), and **GD-Enhancing Tumor** (green). The Tumor Core represents the most aggressive part of the tumor, while Peritumoral Edema indicates the area surrounding the tumor that is affected by edema. The GD-Enhancing Tumor region shows enhancement on Gadolinium (GD) MRI sequences, suggesting a highly vascularized and active part of the tumor.

# Appendix G. BTReport-BraTS: A Companion Dataset for BraTS RRG

Pre-operative mpMRI cases from the combined training and validation splits of the BraTS 2023 Adult Glioma (BraTS'23) dataset (n=1,470 cases) were used to develop the BTReport-BraTS dataset, an open-source companion dataset generated using the BTReport framework. For each case, corresponding midline segmentations, extracted metadata, structured summary reports, and radiology reports generated using BTReport are openly available in the project GitHub page.

## Appendix H. BTReview - A Tool for Radiologist Assessment of Generated Brain Tumor Radiology Reports

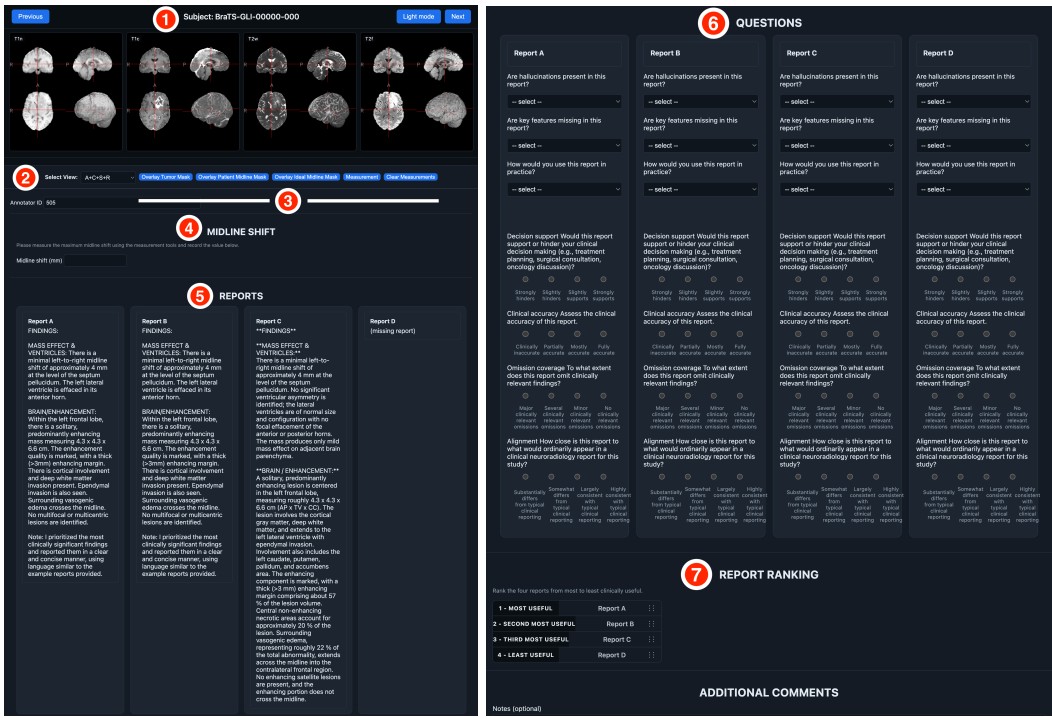

Figure 6: A clinical assessment tool to evaluate the clinical quality of synthetically generated radiology reports generated using various RRG frameworks.

Despite rapid advances in radiology report generation (RRG), there remains a lack of accessible, standardized tools that enable radiologists to efficiently review, compare, and score synthetically generated reports across different RRG frameworks. To address this gap, we developed BTReview, an open-source, web-based platform that supports de-identified case review and double-blinded clinical assessment of generated reports. This platform is customizable with any set of glioblastoma mpMRI cases and is designed to record structured, high-quality radiologist evaluations assessing the overall clinical quality of radiology reports generated with any RRG framework. The generation of these radiologist assessments creates a practical pathway for producing clinically grounded data and error annotation at scale, essential for generating high quality data for VLM training. Furthermore, this validation approach critically evaluates the translatability of existing RRG frameworks to ensure alignment with real-world clinical scenarios. We denote key features of BTReview platform with numerical labels and describe these features below. Feedback from a board-certified neuroradiologist was incorporated into BTReview during the design stage to ensure clinical relevance, appropriate terminology, and alignment with real-world neuroradiology reporting practices.

1) **Interactive Multi-View:** Leveraging the capabilities of NiiVue (Rorden et al., 2024), the first panel is an interactive multi-view of T1n, T1c, T2w, and T2f image sequences that can run on any device (phone, tablet, computer), enabling radiologists to review de-identified mpMRI cases in various clinical environments.

2) **View Selector:** Users can select and modify their multi-view from the drop down menu, enabling them to view image sequences for a given case in the axial, coronal, and sagittal planes, as well as in a 3D render view. An additional option allows all planes to be viewed simultaneously for a given case. Changes in view selection will propagate across all image sequences.

3) **Clinical Toolbar:** Within the clinical toolbar, radiologists can select blue buttons to overlay tumor masks, BTReport-generated midline, and the brain's ideal midline on image sequences in the interactive multi-view. Users can toggle the button again to remove the overlays. Additionally, a measurement tool enables clinicians to obtain quantitative measurements, such as midline shift by dragging their cursor between two points on any image sequence. These measurements will be automatically updated in the multi-view pane, allowing radiologists to revisit measurements across previously annotated cases. Users can also clear their measurements and perform re-measurements if desired.

4) **Midline Shift Measurement:** To validate midline shift measurements generated with BTReport, radiologists can overlay the tumor mask and the brain's ideal midline and scroll to a slice that demonstrates strong midline deviation (e.g., max midline shift). Users can then use the measurement tool to directly obtain a quantitative distance measurement from the boundaries of different sub-regions of the tumor to boundaries demarcated by midline brain structures (e.g., septum pellucidum) for midline shift measurement. Different sub-regions (necrotic core, edema, and enhancing tumor sub-regions) of the tumor mask are color coded and derived from BraTS tumor segmentation. Note: the appearance of tumor segmentation tasks may appear differently (e.g., masks from some institutions may not provide granular parcellation of tumor sub-regions). Radiologists can then overlay the BTReport-generated midline and compare the magnitude of midline deviation through direct measurement and compare their midline shift measurement to the measurement described in synthetically generated radiology reports (see below).

5) **RRG Comparison:** To ensure unbiased evaluation of the clinical quality of synthetically generated radiology reports, the *Findings* description of the reports generated by the BTReport(gpt-oss:120B), BTReport(Llama 3:70B), AutoRG-Brain, and Seg-to-Exp RRG pipelines are displayed, where each column represents the synthetic output generated by a different pipeline. Importantly, synthetically generated reports are described as Report A, B, C, etc., ensuring reviewers are blinded to the identity of the pipeline used to generate the displayed reports. To mitigate potential bias during the review stage, the order in which the radiology report are shown (from left to right) is randomized for each case, ensuring that reviewers are not entrained to any associations with the generated reports and the order in which the reports are displayed.

6) **Clinical Assessment:** After radiologists have had the chance to review image sequences, they will proceed with the clinical assessment, where they will review radiology reports generated through different RRG frameworks. After their review, radiologists will then address a series of questions designed to critique the clinical quality of each generated report based on the following:

a) Generally, the first three questions aim to assess the real-world limitations of synthetic reports generated using automated RRG, and whether radiologists would consider reports clinically useful.

The first question will ask reviewers whether they identified hallucinations across the generated reports. If the radiologist selects "minor" or "major", a conditional question will appear, prompting radiologists to select the types of hallucinations they observed. The next question asks whether key radiological features are missing in this report. Similar to the previous question, if the reviewer selects "some" or "many", a follow up question will appear, prompting them to select from a list of missing elements that would be impactful for improving report quality. The final question asks reviewers to consider how they would personally use each generated report in clinical practice. Here, they will have the opportunity to select from the following responses from the drop down menu: "As a first draft", "As a cross check / second reader", "As a summary aid", and "Would not use."

Summarized below are the questions and their conditional question response options.

| Question | Response Options (single-select) | Conditional Follow-Up (select all that apply) |
|---|---|---|
| **Hallucinations** | None; Minor; Major | *Shown if Minor/Major selected:*

**Type of hallucination(s) observed?**

• Incorrect anatomical location of tumor
• Incorrect tumor characteristics (e.g., size, laterality, enhancement)
• Incorrect clinical implication
• Fabricated finding
• Other (free text: *Other hallucination details*) |
| **Missing Features** | No; Some; Many | *Shown if Some/Many selected:*

**Most impactful missing element(s)?**

• Tumor size/extent
• Enhancement characteristics
• Edema/mass effect
• Midline shift
• Multifocality
• Invasion/eloquent cortex
• Other (*free text*) |
| **Intended use** | As a first draft; As a cross-check/second reader; As a summary aid only; Would not use | — |

Table 7: Clinical evaluation questions and response options, including conditional follow-up items triggered by reviewer selection.

b) Next, radiologists will complete a series of Likert-scale questions asking them to evaluate each generated report based on the following criteria: (a) decision support, (b) clinical accuracy, (c) omission coverage, and (d) clinical structure.

**Decision Support:** Reviewers will be asked to evaluate whether the *Findings* narrative would positively or negatively affect downstream clinical decision making processes (e.g., treatment planning, surgical consult, oncology evaluation).

**Clinical Accuracy:** Reviewers will evaluate the factual accuracy of generated reports.

**Omission Coverage:** Reviewers will evaluate whether clinically relevant findings important for overall radiological interpretation of tumor effects on the surrounding brain environment are included in generated reports, or whether critical clinical findings are lacking or missing.

**Clinical Structure:** The last question asks reviewers to assess the extent to which generated *Findings* narrative differ or align with the clinical organization and conventions typically used in reports.

Summarized below are the question response options for their associated Likert-scale question.

| Question | Criteria | 1 | 2 | 3 | 4 |
|---|---|---|---|---|---|
| Would this report support or hinder your clinical decision making (e.g., treatment planning, surgical consultation, oncology consultation)? | **Decision Support** | Strongly hinders | Slightly hinders | Slightly supports | Strongly supports |
| Assess the clinical accuracy of this report. | **Clinical Accuracy** | Clinically inaccurate | Partially inaccurate | Mostly accurate | Fully accurate |
| To what extent does this report omit clinically relevant findings? | **Clinical Omission** | Major clinically relevant omissions | Several clinically relevant omissions | Minor clinically relevant omissions | No clinically relevant omissions |
| To what extent does this report differ or align with the clinical structure of a standard neuroradiology report? | **Clinical Structure** | Substantially differs from typical clinical reporting | Somewhat differs from typical clinical reporting | Somewhat consistent with typical clinical reporting | Highly consistent with typical clinical reporting |

Table 8: Four-point Likert-scale response options used for radiology report clinical assessment.

7) **Report Ranking:** As a final part of the clinical assessment, radiologists will be asked to rank generated reports based on clinical usefulness. In this question, they will be asked to order reports by "Most useful", "Second most useful", "Third most useful", and "Least useful". At the end of their clinical assessment, radiologists will be given the opportunity to provide optional comments and notes for the case they reviewed before saving their annotation and proceeding to the next case for review. The platform is designed with system memory, meaning that reviewers can revisit previously reviewed cases and make changes to their clinical assessments and annotations. All recorded assessments will be saved as a .json file when the browser closes.

