# OpenReview forum: "BTReport: A Framework for Brain Tumor Radiology Report Generation with Clinically Relevant Features"
_MIDL.io/2026/Conference — MIDL 2026 Poster_

### Official Review · Reviewer_GzBf · 2025-12-16

**Confidence:** 4
**Preliminary Rating:** 2
**Final Rating:** 3

**Summary:**

This work presents BTReport, a framework for brain tumor report generation. Rather than adopting an end-to-end vision–language pipeline, the approach first extracts image-derived features from segmentation, midline shift detection (for which, a novel pipeline is proposed based on clinical guidelines), and VASARI features, which are then provided as inputs to a large language model for report generation. This modular design aims to reduce hallucinations and improve the interpretability of the report generation model.

In addition, the authors introduce BTReport-BraTS, a companion dataset that augments the BraTS imaging benchmark with synthetically generated radiology reports produced using the BTReport framework.

**Strengths:**

- The proposed method is technically solid and presents a well-designed, coherent pipeline.
- Its use of key neuroimaging features, selected based on survival analysis and including aspects such as tumor segmentation and midline shift detection, for report generation is a strong and well-motivated approach.
- Overall, the method has clear potential for real-world application, as it actually mirrors the workflow that radiologists typically follow in clinical practice.

**Weaknesses:**

- The overall organization of the paper can be improved. While the introduction presents the problem and lists the contributions, some elements of the proposed method are described again in the final paragraph of the related work section, followed by a clinical background section that also introduces some methodological details at the end. This makes the narrative difficult to follow before the formal methods section is reached. Furthermore, Methods section start with feature selection methods (Section 5.1), then the overall framework is introduced, then in Section 5.2.4, feature selection is referenced again.
- In Section 5.2.4, the feature assesment criterias are unclear, as both appear to follow the same evaluation procedure. Could authors please clarify whether these assessments are meaningfully distinct or if something is missing?
- The sentence “Following the procedure outlined in Section 5.1.1, each feature was assessed for whether it was a significant predictor of survival time according to the procedure outlined in Section 5.1.2” is unclear and difficult to interpret. How is the procedure outlined in Section 5.1.1 is used exactly? Due to this ambiguity, it is not clear how clustering is used in the method.
- In the Evaluation Metrics section, it is stated that 30 subjects from the UWMC cohort are used, whereas in the Results section, it is mentioned that 25 subjects were evaluated. This inconsistency should be clarified.
- While the limited availability of data is acknowledged, the evaluation of the report generation component is conducted on only (25/30) subjects, which might be insufficient to reliably assess performance using BLEU and ROUGE metrics. Given the high variance of these metrics, conclusions drawn from such a small sample size are not statistically robust. More clinically relevant metrics like manual inspection by a radiologist could give more insights.
- The use of a general-purpose large language model raises questions about whether a domain-specific or medical LLM would be more appropriate, particularly given the clinical nature of the task and the emphasis on reducing hallucinations.
- The claims of two-stage design and the use of neuroimaging-derived features reduce hallucinations can be more strongly supported and discussed in the results section.

**Detailed Comments:**

- VASARI features and abbreviation are referenced several times before being formally introduced; adding a brief explanation at first mention would improve clarity, even if the full description is provided later in the methods.
- Midline shift is used before its abbreviation is defined, and the abbreviation should be introduced at its first occurrence.
- In the introduction, “x-ray” should be written as “X-ray” to follow standard medical capitalization.
- In the related work section, the claim that AutoRG-Brain relies on a vision encoder for quantitative measurements and is therefore prone to hallucinations would benefit from a clearer explanation of why this design choice increases hallucination risk.

**Justification Of Final Rating:**

The authors have adequately addressed my concerns regarding the clarity of the paper. My remaining concern is that the evaluation of the proposed methodology could be extended further. While I appreciate the authors’ effort in presenting new results for intermediate steps, these values should be explicitly compared against appropriate baselines or clinical guidelines to better interpret their significance.

Additionally, as a point of comparison, in an ideal world, it would be interesting to include a direct report generation model to more clearly demonstrate the benefits of BTReport ( which is not currently possible given limited amount of data and makes an another paper by itself).

Overall, I raise my score to 3, with an inclination toward 4.

**Justification Of The Preliminary Rating:**

Overall, I think the method has strong potential; however, the clarity and structure of the paper, as well as the evaluation using more clinically relevant assessments, could be improved to make the work easier to follow and its contributions more compelling.

**Questions To Address In The Rebuttal:**

Please see weaknesses

---

> ### Author Response · Authors · 2026-01-25
> **Rebuttal responses for Reviewer GzBf**
>
> **Reviewer Comment:** *“The overall organization of the paper…some elements of the proposed method are described again in the final paragraph… Furthermore, the Methods section starts with feature selection methods (Section 5.1), then the overall framework is introduced, then in Section 5.2.4, feature selection is referenced again.”*
>
> **Response:** To improve the clarity and organization, we have removed the last paragraph of Section 2: Related Work and incorporated relevant clinical background from Section 3: Clinical Background within the introduction, limiting all descriptions and comparisons of BTReport to Section 4: Methods. Additionally, feature extraction steps described in Section 4.1-4.6: Methods are now reorganized to improve manuscript structure.
>
> **Reviewer Comment:** *“...the feature assessment criteria are unclear… clarify whether these assessments are meaningfully distinct or if something is missing?”*
>
> **Response:** We thank the reviewer for catching the textual duplication in the initial submission and apologize for any confusion it may have caused during the review process. This section is now clarified in the resubmission.
>
> **Reviewer Comment:** *“The sentence… is unclear and difficult to interpret... Due to this ambiguity, it is not clear how clustering is used in the method.”*
>
> **Response:** We agree that the sentence is ambiguous and have revised it for clarity. Semantic clustering was used to establish a list of higher-ordered clinical concepts described in radiology reports across all cases in the HuskyBrain dataset. Distilling the top 35 most-frequently mentioned radiology concepts from radiologist-authored reports, as shown in Appendix A, was pivotal for evaluating the clinical relevance of BTReport features, not feature selection.
>
> **Reviewer Comment:** *“...the evaluation of the report generation component is conducted on only (25/30) subjects, which might be insufficient to reliably assess performance… clinically relevant metrics like manual inspection by a radiologist could give more insights.” && “It is stated that 30 subjects… whereas in the Results section, it is mentioned that 25 subjects were evaluated. This inconsistency should be clarified.”*
>
> **Response:** We have addressed this concern by adding 70 more cases to the validation, to total 100 subjects. This change is now reflected in the revised manuscript.
>
> **Reviewer Comment:** *“...use of a general-purpose large language model raises questions about whether a domain-specific or medical LLM would be more appropriate…”*
>
> **Response:** We thank the reviewer for this comment. In the BTReport framework, the LLM is not used for medical image interpretation, clinical inference, or medical decision support. Rather, its sole purpose is for syntactic structuring and narrative synthesis of pre-extracted, deterministic clinical features. Therefore, our priority was to select LLMs which have strong instruction-following capabilities, such as gpt-oss:120b and llama3:70b. Experiments with domain-specific models like MedGemma were unsuccessful, as the models struggled with instruction following. To this end, we did not include these in the manuscript.
>
> **Reviewer Comment:** *“The claims of two-stage design and the use of neuroimaging-derived features to reduce hallucinations can be more strongly supported and discussed in the results section.”*
>
> **Response:** We agree with the reviewer that our claims of existing RRG frameworks being more susceptible to hallucinations was poorly supported. As descriptions of model hallucination encompass many different task domains, we decided to remove all hallucination claims, and have rewritten the related works section.
>
> **Reviewer Comment:** *“VASARI features and abbreviation are referenced several times before being formally introduced; adding a brief explanation at first mention would improve clarity…”*
>
> **Response:** We agree with the reviewer about including a brief explanation of what VASARI features are and have revised the second to last paragraph of Section 1: Introduction.
>
> **Reviewer Comment:** *“Midline shift ...should be introduced at its first occurrence.”*
>
> **Response:** We appreciate the grammatical suggestion. This is now revised in the appropriate sections in the resubmission.
>
> **Reviewer Comment:** *“...”x-ray” should be written as “X-ray”...”*
>
> **Response:** We appreciate the suggestion of following standard medical capitalization. This is now revised in the appropriate sections in the resubmission.
>
> **Reviewer Comment:** *“In the related works section… AutoRG-Brain relies on a vision encoder for quantitative measurements and is therefore prone to hallucinations would benefit from a clearer explanation…”*
>
> **Response:** We agree with this suggestion and have rewritten this section to confine our claims of AutoRG-Brain strengths and limitations based on (1) methodological design, and (2) clinical report quality. In general, related work has been rewritten using this criteria.

---

> > ### Comment · Reviewer_GzBf · 2026-01-26
> >
> > I thank the authors for their response. The methodology is now presented more clearly and paper reads more easily.
> >
> > However, I still believe that a more detailed evaluation would strengthen the paper. Since the pipeline consists of multiple stages (e.g., segmentation), the performance of each step could be evaluated individually. Currently, the results are assessed only using LLM-based metrics, where LLM is primarily intended to summarize the outputs of the preceding stages and does not add additional information. So if the previous stages are performing well, LLMs are expected to create a report from the findings.

---

> > > ### Author Response · Authors · 2026-01-29
> > > **New Section: Reliability of Extracted Features**
> > >
> > > Thank you for this comment. We agree that understanding the reliability of extracted features is important, as this will directly impact the quality of reports downstream.
> > >
> > > We have added a section "Reliability of Extracted Features" to the paper, where we validate the intermediate features with labels extracted from radiologist prompts. For example, many reports include midline shift values, so we extract these using Google's LangExtract and compare with the predictions from BTReport. This procedure was repeated for multiple features and we report accuracy for binary/categorical features like side of tumor epicenter, and MAE for continuous features like tumor volume.

---

### Official Review · Reviewer_6PkM · 2025-12-18

**Confidence:** 4
**Preliminary Rating:** 4
**Final Rating:** 4

**Summary:**

The paper proposes a method called BTReport that generates radiological reports for brain tumour patients using clinically relevant features, such as tumor volume, instead of traditional methods that rely mainly on the input images. The work is validated on public data, and the results are compared against other methods in the literature (AutoRG-Brain, Seg-to-Exp.)

**Strengths:**

The paper is well-written and relevant, as mitigating hallucinations in radiology report generation is critical to the translation of this technology. The work proposes an interesting framework for generating the radiology reports of the brain tumor, specifically glioblastoma patients.

The work is reproducible, with the code and data used in the experiments being publicly available.

**Weaknesses:**

I would imagine the proposed method is highly dependent on the quality/accuracy/robustness of the extracted clinical features. For example, tumor segmentation, brain segmentation, and midline shift analysis methods are imperfect. How would this affect the performance of the model? Was there any quality control of the features extracted?

I missed details about how the author came up with the BTReport prompt. For example, rule #4 "Never mention imaging sequences other than T1n, T2w, T2 FLAIR, or T1-Gd. Do not mention diffusion, perfusion, spectroscopy, MRA, or other modalities (unless stated explicitly in metadata)" is very curious. Without it, was your model hallucinating reports that included non-existing sequences?

I missed evaluation of the results by medical experts

**Detailed Comments:**

- To the best of my knowledge, TextBraTS used only the FLAIR sequence and not all MRI sequences to generate their radiology reports.

-   It would be good to report not just the metric mean or median, it is unclear from the table, but also the standard deviation.

- The model doesn't require training? If so, this should be emphasized.

- The authors mention a novel dataset with n=1,470 cases, but they only tested the model on 25 or 30 (evaluationmetrics and results say different things) samples? Why?

**Justification Of Final Rating:**

Thank you to the authors for their detailed response. My remaining concern is the same as reviewer GzBf and relates to my comment that the model depends on the quality of the extracted clinical features. Although the methods used are published and open-source, they are not a clinical standard and often fail. A better understanding of error propagation is critical for a better understanding of the feasibility and robustness of the proposed method.

**Justification Of The Preliminary Rating:**

Although some points could be clarified in the paper, this is interesting work that could lead to improved automated radiology report generation. The fact that the code is publicly available and the authors are releasing new data linked to BRATS 2023 is a nice bonus.

**Questions To Address In The Rebuttal:**

- Please include details about robustness of the method to errors in in the clinical features being used as input to the model
- Please clarify the data and how it was used to validate the model
- Please discuss in more detail the generation of the prompt use din the work

---

> ### Author Response · Authors · 2026-01-25
> **Rebuttal responses for Reviewer 6PkM**
>
> **Reviewer Comment:**
> *“...proposed method is highly dependent on the quality/accuracy/robustness of the extracted clinical features… How would this affect the performance of the model? Was there any quality control of the features extracted?”*
>
> **Response:**
> While we agree with reviewers that consistent RRG depends on the robustness of extracted clinical features, we believe that the open-sourced pipelines used in the BTReport feature extraction stage are robust, as their methods are largely supported, published, and have been extensively peer-reviewed.
>
> **Reviewer Comment:**
> *“(1)...how the author came up with the BTReport prompt… rule #4… is very curious. Without it, was your model hallucinating reports that included non-existing sequences?” (2) “Please discuss in more detail the generation of the prompt used in the work.”*
>
>  **Response:**
> We acknowledge the reviewer’s curiosity with how the BTReport prompt was generated. Our motivation for generating the LLM prompt in Appendix C with a highly-specific ruleset was to maximize the LLM’s capability to efficiently perform in-context learning and narrative synthesis. Since the BTReport framework is currently limited to standard mpMRI imaging modalities (e.g., T1n, T2w, T2 FLAIR, and T1-Gd) for deterministic feature extraction, ground truth radiology findings that report findings across other MRI modalities (e.g., diffusion weighted imaging, MRA/perfusion imaging, spectroscopy) were out of the scope of the LLM’s capabilities for RRG. This is well illustrated in Appendix D, where the ground truth radiology report describes “restricted diffusion” and “microhemorrhages/angiovision.” These clinical findings (emphasized with **) are derived from diffusion weighted imaging and perfusion-based imaging. As the second stage of narrative synthesis using LLM-based RRG is constrained solely to the deterministic features extracted from existing sequences, there is no way for the LLM to ascribe observations from non-existing sequences to downstream RRG. Importantly, when the rule #4 prompt was revised to remove the highly structured inclusion/exclusion criteria for imaging sequences, there were observed incidents of LLM hallucination of findings from non-existing imaging sequences.
>
> **Reviewer Comment:**
> *“I missed evaluation of the results by medical experts”*
>
> **Response:**
> Though we did not initially consider generating a new radiologist validation set, we propose a systematic framework for large-scale radiologist assessment of synthetically generated radiology reports (discussed in newly added Appendix F).
>
> **Reviewer Comment:**
>  *“...TextBraTS used only the FLAIR sequence and not all MRI sequences…”*
>
> **Response:**
>  The authors in the TextBraTS study (Xiaoyu et al., 2025) used BraTS cases with T1, T1-Gd, T2, and FLAIR sequences. However, they specifically mentioned that the tumor segmentation annotations used for their textual descriptions were derived from the FLAIR sequence. As a result, the reviewer is correct. We have added this detail in the revised related work.
>
> **Reviewer Comment:**
> *“It would be good to report not just the metric mean or median,... but also the standard deviation.”*
>
> **Response:**
>  We agree with the reviewer’s comment and have added the standard deviation to mean automated metrics in Tables 2 and 3.
>
> **Reviewer Comment:**
>  *“The model doesn’t require training? If so, this should be emphasized.”*
>
> **Response:** The reviewer comment is correct in that the LLM model does not require further training. We have further emphasized this in the revised manuscript.
>
> **Reviewer Comment:**
> *“...a novel dataset with n=1,470 cases, but they only tested the model on 25 or 30... Why?”*
>
> **Response:** The BTReport-BraTS dataset was generated by performing inference with BTReport using unlabeled cases from the BraTS’23 dataset. While we produce companion BTReport-generated reports, the BraTS’23 dataset did not have ground truth radiology reports, limiting our ability to the BTReport-BraTS dataset. In contrast, our evaluation dataset, HuskyBrain, is much smaller (n=155), but contains corresponding radiologist-authored reports.
>
> **Reviewer Comment:**
> *(1) “Please include details about robustness of the method to errors in the clinical features being used as input to the model.” (2) “Please clarify the data and how it was used to validate the model.”*
>
> **Response:** While we agree with reviewers that consistent RRG depends on the robustness of extracted clinical features, we believe that the open-sourced pipelines used in the BTReport feature extraction stage are robust, as their methods are largely supported, published, and have been extensively peer-reviewed. However, midline shift measurement, a quantitative feature generated with the methodology proposed in this study, remains unvalidated. To address this, we have taken the proper steps to ensure measurements are validated with radiologist measurements using BTReview.

---

> > ### Comment · Reviewer_6PkM · 2026-01-27
> >
> > Thank you for the authors' detailed response. My remaining concern is the same as reviewer GzBf and relates to my comment that the model depends on the quality of the extracted clinical features.  Though the methods used are published, open-source methods, they are not a clinical standard and often fail. A better understanding of error propagation is critical.

---

> > > ### Author Response · Authors · 2026-02-02
> > > **Evalaution of extracted feature quality and error propagation**
> > >
> > > We thank the reviewers for this comment, and agree that additional validation of  BTReport extracted features and understanding error propagation is critical. In response, we have conducted two further experiments.
> > >
> > > 1. We have added a section "Reliability of Extracted Features" to the paper, where we evaluate the reliability of BTReport-extracted clinical features by comparing them with the same features extracted from ground-truth radiology reports in the HuskyBrain dataset.  For example, many reports include midline shift values, so we extract these using Google's LangExtract and compare with the predictions from BTReport. This procedure was repeated for multiple features and we report accuracy for binary/categorical features like side of tumor epicenter, and MAE for continuous features like tumor volume.
> > >
> > > 2. To evaluate the effects of error propagation resulting from inadequate segmentation masks, we have added additional experiments to all of our results tables. Here, we evaluate how the outputs (features and reports) of BTReport differ when we use an inferior segmentation algorithm (DeepMedic [1]; dual-pathway
> > > CNN), from when we use a superior algorithm (Faking It [2]; winners of BraTS'23 challenge, uses an ensemble of nnU-Nets).
> > >
> > > Updated results tables for both of these additions are shown below. We find that improved segmentations help with  feature quality, especially with number of lesions; lexical metrics remain largely unaffected; and factual metrics significantly improve. If possible, we would like to add these results to the final version of the paper as they significantly strengthen the discussion.
> > >
> > > [1]: https://www.sciencedirect.com/science/article/pii/S1361841516301839
> > > [2]: https://arxiv.org/abs/2402.17317
> > >
> > >
> > > ---
> > >
> > > ### Reliability of Extracted Features
> > >
> > > | Method         | Cortical involvement | Side of tumor epicenter | Ventricular effacement | Midline shift (mm) | Number of lesions | Tumor volume (cm³) |
> > > |---------------|----------------------|--------------------------|------------------------|-------------------|-------------------|--------------------|
> > > | BTReport (V1) | 0.79                 | **1.00**                 | **0.56**               | 1.26 ± 2.60       | 1.48 ± 2.63       | **1.80 ± 3.95**    |
> > > | BTReport (V2) | **0.80**             | **1.00**                 | 0.52                   | **1.21 ± 2.55**   | **0.41 ± 1.69**   | 2.00 ± 5.16        |
> > >
> > > V1: DeepMedic segmentations
> > > V2: Faking It segmentations
> > >
> > >
> > > ---
> > > ### Lexical Metrics
> > >
> > > Mean ± standard deviation BLEU and ROUGE metrics.
> > >
> > > | Framework                         | BLEU-1            | BLEU-2            | ROUGE-1           | ROUGE-2           |
> > > |-----------------------------------|-------------------|-------------------|-------------------|-------------------|
> > > | BTReport (gpt-oss:120B, V1)†      | 0.236 ± 0.068     | 0.124 ± 0.044     | 0.360 ± 0.078     | 0.109 ± 0.038     |
> > > | BTReport (gpt-oss:120B, V2)†      | 0.244 ± 0.070     | 0.132 ± 0.044     | **0.371 ± 0.078** | **0.115 ± 0.040** |
> > > | BTReport (LLaMA3:70B, V1)†        | **0.248 ± 0.078** | **0.136 ± 0.052** | 0.362 ± 0.080     | **0.115 ± 0.043** |
> > > | AutoRG-Brain                      | 0.158 ± 0.072     | 0.080 ± 0.042     | 0.268 ± 0.060     | 0.070 ± 0.032     |
> > > | Seg-to-Exp                        | 0.085 ± 0.039     | 0.035 ± 0.018     | 0.163 ± 0.055     | 0.023 ± 0.013     |
> > >
> > > † Following Approximate Randomization, BTReport variants outperform other frameworks across all metrics.
> > >
> > > V1: DeepMedic segmentations
> > > V2: Faking It segmentations
> > >
> > > ---
> > > ### Factual Accuracy Metrics
> > >
> > > Mean ± standard deviation TBFact, BERTScore, and RaTEScore metrics.
> > >
> > > | Framework                         | TBFact Score       | TBFact Prec.      | TBFact Recall     | TBFact F1         | BERTScore         | RaTEScore         |
> > > |-----------------------------------|--------------------|-------------------|-------------------|-------------------|-------------------|-------------------|
> > > | BTReport (gpt-oss:120B, V1)†      | 0.313 ± 0.145      | 0.345 ± 0.155     | 0.325 ± 0.163     | 0.313 ± 0.145     | 0.447 ± 0.060     | 0.568 ± 0.057     |
> > > | BTReport (gpt-oss:120B, V2)†      | **0.353 ± 0.151**  | **0.412 ± 0.146** | **0.349 ± 0.162** | **0.359 ± 0.145** | **0.453 ± 0.055** | **0.577 ± 0.054** |
> > > | BTReport (LLaMA3:70B, V1)†        | 0.295 ± 0.130      | 0.377 ± 0.168     | 0.274 ± 0.136     | 0.295 ± 0.130     | 0.433 ± 0.062     | 0.567 ± 0.061     |
> > > | AutoRG-Brain                      | 0.072 ± 0.123      | 0.282 ± 0.155     | 0.186 ± 0.137     | 0.196 ± 0.130     | 0.327 ± 0.047     | 0.477 ± 0.053     |
> > > | Seg-to-Exp                        | 0.014 ± 0.047      | 0.131 ± 0.121     | 0.147 ± 0.108     | 0.098 ± 0.089     | 0.156 ± 0.042     | 0.409 ± 0.039     |
> > >
> > > V1: DeepMedic segmentations
> > > V2: Faking It segmentations

---

### Official Review · Reviewer_UHta · 2026-01-15

**Confidence:** 4
**Preliminary Rating:** 4
**Final Rating:** 4

**Summary:**

This paper presents BTReport, a novel two-stage framework for generating radiology reports from brain tumor MRI that effectively mitigates hallucination and omission issues prevalent in end-to-end vision-language models. It utilizes quantitative imaging features, validated for clinical relevance, to constrain a general-purpose LLM via structured prompting, thereby reducing hallucinations. The framework demonstrates superior factual accuracy compared to existing baselines and contributes an open-source dataset to advance the field.

**Strengths:**

This paper presents a significant methodological contribution through its well-designed two-stage framework, which effectively mitigates key limitations of existing vision-language models in radiology report generation, notably hallucination and omission errors. By decoupling deterministic, interpretable feature extraction, including robust 3D midline shift quantification and VASARI-derived anatomical descriptors, from a prompting-based LLM narrative synthesis, the authors establish a reliable, clinically grounded pipeline that enhances transparency and factual consistency. Notably, the approach does not require domain-specific fine-tuning, instead leveraging constrained in-context learning to align report style with clinical standards while utilizing features validated for prognostic relevance via survival analysis. The rigorous evaluation using factuality-centered metrics and the release of an open-source dataset further strengthen the paper's reproducibility and potential value to the neuroimaging community.

**Weaknesses:**

While the paper presents a well-structured and promising framework, several methodological and experimental limitations warrant consideration.

First, the framework's reliance on a cascade of external algorithms for segmentation and registration introduces risks of error propagation, where inaccuracies in upstream components could systematically affect downstream features and the final report.

Second, the computational efficiency (i.e. the runtime from image input to report generation) of the full pipeline, from feature extraction to LLM-based generation, is not reported, limiting assessment of its potential for real-time clinical use.

Third, evaluation is confined to 30 glioblastoma cases from a single institution; performance across diverse tumor types, imaging protocols, or patient populations remains unvalidated.

Additionally, the LLM functions strictly as a constrained narrative synthesizer, precluding it from performing higher-level clinical reasoning, such as integrating multi-modal features to suggest diagnostic probabilities or prognostic implications, that could enhance decision support.

Finally, the clinical utility of generated reports lacks radiologist assessment, leaving their practical diagnostic reliability unclear.

Addressing these aspects would enhance the framework's robustness and translational relevance.

**Detailed Comments:**

1. The paper would benefit from reporting the total inference time of the BTReport pipeline, from image input to final generated report, and comparing it with baseline models. Additionally, reporting the model size of  baseline methods would offer a more balanced view of the frameworks' architectural complexity.

2.  To strengthen the evaluation of clinical utility beyond automated metrics, it is suggested to incorporate a reader study where board-certified neuroradiologists assess the synthetic reports. Evaluating aspects such as diagnostic confidence, completeness, and potential for integration into the clinical workflow would provide direct evidence of real-world applicability.

3. There is a textual duplication in the description of the two criteria for feature selection in Section 5.2.4. The second criterion currently repeats the first. This should be revised for clarity, presumably to distinguish between "clinical report frequency" and "prognostic significance," as implied by the surrounding context.

4. In Section 5.2.5, consider adding a brief note to explicitly state that the LLM's role is confined to narrative synthesis based on provided features and does not encompass independent medical reasoning or inference from the images. This would further accentuate the conceptual distinction from end-to-end VLM approaches.

**Justification Of Final Rating:**

The BTReport framework makes a compelling methodological contribution by introducing a two-stage, feature-driven paradigm for radiology report generation, which effectively addresses the critical issue of hallucination in neural report synthesis. The design is clinically grounded, interpretable, and well-executed, supported by quantitative validation using survival-relevant features and rigorous automated metrics. While the study has limitations, particularly in evaluation scope and clinical validation, its strengths in novelty, transparency, and reproducibility outweigh these shortcomings. With appropriate revisions and responses to reviewer feedback, this work represents a meaningful advance toward reliable AI-assisted reporting in neuro-oncology.

**Justification Of The Preliminary Rating:**

The BTReport framework makes a compelling methodological contribution by introducing a two-stage, feature-driven paradigm for radiology report generation, which effectively addresses the critical issue of hallucination in neural report synthesis. The design is clinically grounded, interpretable, and well-executed, supported by quantitative validation using survival-relevant features and rigorous automated metrics. While the study has limitations, particularly in evaluation scope and clinical validation, its strengths in novelty, transparency, and reproducibility outweigh these shortcomings. With appropriate revisions and responses to reviewer feedback, this work represents a meaningful advance toward reliable AI-assisted reporting in neuro-oncology.

**Questions To Address In The Rebuttal:**

Please refer to the points raised in the Detailed Comments.

---

> ### Author Response · Authors · 2026-01-25
> **Responses to Reviewer UHta**
>
> **Reviewer Comment:** *“...introduces risks of error propagation…inaccuracies in upstream components could systematically affect downstream features and the final report.”*
>
> **Response:** We thank the reviewer for raising this comment. While cascaded pipelines are susceptible to error propagation, BTReport is intentionally designed to mitigate this risk. All upstream steps in BTReport perform deterministic, task-specific operations, ensuring that errors remain localized and inspectable rather than being implicitly absorbed by downstream steps.
> All extracted features are explicitly computed, logged, and exposed to the user prior to report generation, enabling attributable and transparent auditing. BTReport relies on validated neuroimaging algorithms robust to pathological anatomy, including SynthSeg and SynthMorph, which were selected for their resilience to tumor-induced anatomical distortions, reducing systemic failure relative to general approaches (e.g., FreeSurfer) where the deformation and registration steps perform poorly. To further mitigate risk of error propagation, downstream radiology report generation (RRG) is feature-constrained by design: LLM narrative synthesis is restricted to only extracted imaging-derived features, and prohibited from inferring unsupported findings, preventing error compounding. This modularity mirrors standard clinical workflows unlike end-to-end-vision-language models, enabling stage-wise traceable error attribution and framework improvement.
>
> **Reviewer Comment:** *(1) “...computational efficiency (i.e., the runtime from image input to report generation…is not reported,...” (2) “...reporting the total inference time… from image input to final generated report, and comparing it with baseline models… reporting the model size of baseline methods would offer a more balanced view…”*
>
> **Response:** We agree that evaluating the computational efficiency of BTReport pipeline is a notable pre-requisite for its clinical adoption. BTReport was designed to mirror offline radiology reporting workflows rather than real-time deployment. The pipeline decouples deterministic feature extraction from LLM-based RRG, enabling computationally intensive steps (e.g., segmentation and registration) to be executed asynchronously. These upstream feature extraction steps can be efficiently parallelized across distributed high performance compute clusters or institutional GPU infrastructure, enabling high-throughput processing at scale. Once features are extracted, computational cost for LLM-based RRG runtime is negligible by comparison (seconds). We have clarified these design considerations, but did not have adequate time within the seven day rebuttal period to provide robust runtime metrics for BTReport-processed cases. We recognize the importance of these metrics for evaluation for clinical implementation and plan to address this in future updates to the BTReport framework.
>
> **Reviewer Comment:** *“...confined to 30 glioblastoma cases from a single institution; performance across diverse tumor types, imaging protocols, or patient populations remains unvalidated.”*
>
> **Response:** We recognize the reviewer’s concern about the limited sample size for automated metric evaluation. During the rebuttal period, we have added an additional 70 cases (100 total cases) to the HuskyBrain dataset for automated metric comparison.
>
> **Reviewer Comment:** *(1) “...LLM functions strictly as a constrained narrative synthesizer…that could enhance decision support.” (2) “In Section 5.2.5, consider adding a brief note to explicitly state that the LLM’s role is confined to narrative synthesis…”*
>
> **Response:** We agree with the reviewer that the role of the LLM for RRG needs to be better described. This is now reflected in the revised manuscript.
>
> **Reviewer Comment:** *(1) “...lacks radiologist assessment, leaving their practical diagnostic reliability unclear.” (2) “...incorporate a reader study where board-certified neuroradiologists assess the synthetic reports…”*
>
> **Response:** We agree that the lack of clinical assessment of synthetically generated radiology reports from BTReport poses a central limitation of the proposed BTReport framework. We have taken the reviewer’s suggestion for validating BTReport with a reader study and designed BTReview, a double-blinded assessment tool that enables fellowship-trained radiologists to evaluate the clinical quality of reports generated using BTReport framework and other comparison RRG models. The platform provides de-identified glioblastoma cases that neuroradiologists can interactively review, and provides structured clinical assessment questions to evaluate the clinical quality of generated reports.
>
> **Reviewer Comment:** *“...textual duplication…in Section 5.2.4…”*
>
> **Response:** We thank the reviewer for catching this typographical error and apologize for any confusion it may have caused during the review process. This is now addressed in the revised manuscript.

---

### Author Rebuttal · Authors · 2026-01-25

**Rebuttal:**

The authors thank the MIDL Editors and Reviewers for the opportunity to revise the manuscript. We made targeted revisions to strengthen our claims and clarify our contributions. First, our statement that existing neuro-oncology RRG frameworks are more prone to hallucinations was insufficiently supported. We removed these claims and rewrote the related work to focus on (1) methodological design, and (2) clinical report quality.

In re-assessing RRG outputs against radiologist-authored reports, we identified a key distinction not clearly articulated in the initial submission: BTReport generates reports from clinically relevant quantitative features. Baseline models rarely have both quantitative measurement features and radiology report-relevant features. This leads to generated reports that are not clinically meaningful and diverge from reference report structure, lexical similarity, and factual accuracy. Based on these observations, we reframed BTReport as a measurement-grounded RRG framework. BTReport performs RRG solely from deterministically extracted features, enabling fully textual report generation without reliance on vision encoders/VLMs for measurement operations. These distinctions are now emphasized in the revised Methodology.

Upon further review, we also revised the description of feature selection. Semantic clustering and survival modeling were previously described as supporting RRG, but their purpose was unclear. We now present these analyses as feature validation steps used to verify the clinical relevance of BTReport features.

A recurring reviewer concern was the lack of clinical validation of BTReport-generated reports, including unvalidated automated midline shift measurements. We agree these are central limitations affecting clinical translation. Given the rebuttal timeline, collecting a new radiologist-annotated validation set was not feasible. To address this gap, we designed BTReview, a double-blinded radiologist assessment protocol for BTReport and baseline RRG frameworks. BTReview evaluates midline shift measurements and overall report quality across four criteria: clinical accuracy, omission coverage, and clinical structure.

Other revisions include expanding automated metrics evaluation to 100 HuskyBrain cases. Together, we hope these revisions adequately address reviewer concerns and improve the clarity and positioning of BTReport.

Zip file contains: Revised manuscript and manuscript with tracked changes.

**Supporting Material:**

/attachment/327aa5003968c6254dbbd6c8526f24c593cbb485.zip

---

### Meta-Review · Area_Chair_iEt3 · 2026-02-09

**Recommendation:** Accept (Poster)
**Confidence:** 5

**Metareview:**

The paper is recommended for ACCEPT based on the overall positive consensus and the strength of the contribution. Reviewers 1 and 2 both provided "weak accept" ratings, highlighting that the proposed BTReport framework is a clinically grounded methodological advance that improves interpretability and reduces hallucinations through a two-stage design. Reviewer 3 is borderline but noted that the rebuttal successfully clarified several issues and subsequently raised their score.

---

### Decision · Program_Chairs · 2026-02-13

Accept (Poster)